# Dapagliflozin Prevents NOX- and SGLT2-Dependent Oxidative Stress in Lens Cells Exposed to Fructose-Induced Diabetes Mellitus

**DOI:** 10.3390/ijms20184357

**Published:** 2019-09-05

**Authors:** Ying-Ying Chen, Tsung-Tien Wu, Chiu-Yi Ho, Tung-Chen Yeh, Gwo-Ching Sun, Ya-Hsin Kung, Tzyy-Yue Wong, Ching-Jiunn Tseng, Pei-Wen Cheng

**Affiliations:** 1Department of Ophthalmology, Kaohsiung Veterans General Hospital, Kaohsiung 81362, Taiwan (Y.-Y.C.) (T.-T.W.) (Y.-H.K.); 2School of Medicine, National Yang-Ming University, Taipei 11221, Taiwan; 3Department of Medical Education and Research, Kaohsiung Veterans General Hospital, Kaohsiung 81362, Taiwan (C.-Y.H.) (T.-Y.W.); 4Department of Biomedical Sciences, National Sun Yat-Sen University, Kaohsiung 80424, Taiwan; 5Department of Internal Medicine, Division of Cardiology, Kaohsiung Veterans General Hospital, Kaohsiung 81362, Taiwan; 6Department of Anesthesiology, Kaohsiung Medical University Hospital, Kaohsiung Medical University, Kaohsiung 80708, Taiwan; 7Department of Pharmacology, National Defense Medical Center, Taipei 11490, Taiwan; 8Department of Medical Research, China Medical University Hospital, China Medical University, Taichung 40402, Taiwan

**Keywords:** cataract, dapagliflozin, type 2 diabetes mellitus, NADPH oxidase, glucose transporter

## Abstract

Purpose: Cataracts in patients with diabetes mellitus (DM) are a major cause of blindness in developed and developing countries. This study aims to examine whether the generation of reactive oxygen species (ROS) via the increased expression of glucose transporters (GLUTs) and the receptor for advanced glycation end products (RAGE) influences the cataract development in DM. Methods: Lens epithelial cells (LECs) were isolated during cataract surgery from patients without DM or with DM, but without diabetic retinopathy. In a rat model, fructose (10% fructose, 8 or 12 weeks) with or without dapagliflozin (1.2 mg/day, 2 weeks) treatment did induce DM, as verified by blood pressure and serum parameter measurements. Immunofluorescence stainings and immunoblottings were used to quantify the protein levels. Endogenous O_2_˙¯ production in the LECs was determined in vivo with dihydroethidium stainings. Results: We investigated that GLUT levels in LECs differed significantly, thus leading to the direct enhancement of RAGE-associated superoxide generation in DM patients with cataracts. Superoxide production was significantly higher in LECs from rats with fructose-induced type 2 DM, whereas treatment with the sodium-glucose cotransporter 2 (SGLT2) inhibitor dapagliflozin prevented this effect in fructose-fed rats. Protein expression levels of the sodium/glucose cotransporter 2 (SGLT2), GLUT1, GLUT5, the nicotinamide adenine dinucleotide phosphate reduced form (NADPH) oxidase subunit p67-phox, NOX2/4 and RAGE were upregulated in fructose-fed animals, whereas dapagliflozin treatment reversed these effects. Conclusions: In rats with fructose-induced DM, dapagliflozin downregulates RAGE-induced NADPH oxidase expression in LECs via the inactivation of GLUTs and a reduction in ROS generation. These novel findings suggest that the SGLT2 inhibitor dapagliflozin may be a candidate for the pharmacological prevention of cataracts in patients with DM.

## 1. Introduction

In late 2014, the United Nations (UN) declared that type 2 diabetes mellitus (DM), obesity and metabolic syndrome had outpaced infectious diseases as the main global threat to human health [1]. The World Health Organization (WHO) highlighted high fructose consumption, mainly in the form of sweetened beverages, as a risk factor for several metabolic diseases [2]. DM is a systemic disease that affects many organs, including the eye, and leads to high healthcare costs. Patients with DM also tend to develop cataracts earlier in life than those without DM. Cataracts are a key factor in diabetes-associated morbidity, although the pathogenesis of diabetic cataracts is poorly understood [3].

Previous research has demonstrated that the chronic hyperglycemia-induced overproduction of reactive oxygen species (ROS) plays a central role in the pathogenesis of diabetic complications, including diabetic cataracts [4]. High ROS levels directly disturb the physiological functions of cellular macromolecules and subsequently lead to lens opacification. Currently, surgery for cataract removal and intraocular lens implants are the main treatments for diabetic cataracts. However, surgery may result in severe postoperative complications, including infection, corneal edema, and ocular hypertension, especially in the elderly, and those with hyperglycemic conditions [5]. Therefore, additional therapeutic strategies must be developed for the prevention and treatment of diabetic cataracts.

Model animal species develop metabolic syndrome after several weeks of high fructose consumption, characterized by insulin resistance, impaired glucose homeostasis, and hypertension [6]. There is accumulating evidence that fructose promotes an oxidative imbalance by simultaneously enhancing ROS production and downregulating key antioxidant enzymes, such as superoxide dismutase (SOD) [7,8]. The phagocyte-type nicotinamide adenine dinucleotide phosphate (NADPH) oxidase has recently been identified as the major source of ROS in the vasculature [9]. This NADPH oxidase is composed of two catalytic (gp91-phox and p22-phox) and four regulatory (p47-phox, p40-phox, p67-phox and Ras-related C3 botulinum toxin substrate 1 (Rac1)) subunits. Furthermore, the dietary-fructose-mediated generation of advanced glycation end products (AGEs) and the activation of the receptor for AGEs (RAGE), both of which are senescent protein derivatives that result from the auto-oxidation of glucose and fructose, contribute to metabolic syndrome [10,11]. Additionally, AGEs and their receptor, RAGE, may directly induce the generation of ROS via NADPH oxidases or other identified mechanisms [12].

Fructose is absorbed via glucose transporters (GLUTs) in the intestine (mainly GLUT2 and GLUT5). Recent research has focused upon GLUT5 and its regulation [13]. Mantych et al. suggested that GLUT1 is the main glucose transporter, and is typically localized in lens epithelial cells (LECs), a part of the blood-aqueous barrier. However, GLUT5 is the main fructose transporter of the human eye [14]. Sodium-dependent glucose cotransporter 2 (SGLT2) inhibitors have recently been developed for glycemic control in patients with type 2 DM [15]. Lou et al. demonstrates that the effect of pioglitazone on insulin resistance in fructose-drinking rats correlates with AGEs/RAGE inhibition and blocking the activation of NADPH oxidase and NF kappa B [16]. According to immunofluorescence staining, GLUT1 and SGLT2 are highly expressed in both human and murine LECs [17]. LECs also play a crucial role in antioxidant protection and the transport of nutrients from the aqueous humor. However, the effects of GLUTs and Rac1-dependent ROS in human LECs on the regulation of fructose-induced diabetic cataracts in vivo or in vitro are unknown.

We hypothesized that the use of an SGLT inhibitor would decrease RAGE-induced NADPH oxidase expression, thereby reducing ROS generation during the formation of fructose-induced diabetic cataracts.

## 2. Results

### 2.1. GLUTs May Act Through RAGE and Induce RAC1 Expression and Superoxide Production in the Cataract Lenses of DM Patients

During aging and in chronic hyperglycemia, the accumulation of oxidized lens components and the decreased efficiency of repair mechanisms can contribute to the development of lens opacities or cataracts [18,19]. We enrolled 30 DM (+) patients with cataracts as the study group (15 males, 15 females) and 30 DM (−) patients with cataracts as the control group (18 males, 12 females). Refraction and axial length were assessed in the DM (15 and 15 patients, respectively) and the control (18 and 12 patients, respectively) groups. The average age was 62.0 ± 9.5 years for the cataract patients and 62.3 ± 7.0 years for the DM patients (*p* = 0.2). A significant difference in the HbA1c levels was found between the study and the control groups (Table 1). We also investigated whether GLUT levels in the LECs differed significantly, thus leading to a direct enhancement of RAGE-associated superoxide generation in the DM patients with cataracts. Immunofluorescence staining determined the SGLT2 and 3-nitrotyrosine (3-NT) levels in the LECs of patients with and without DM. Representative images are shown in Figure 1A. The results show that SGLT2 levels were significantly higher in the DM patient LECs. However, high 3-NT levels were observed in those LECs of patients with and without DM. Interestingly, immunofluorescence staining demonstrated that RAGE and GLUT1 levels were significantly higher in the LECs of patients with DM (Figure 1B,C).

### 2.2. Dapagliflozin Prevents Fructose-Mediated Metabolic Defects

Table 2 provides the blood pressure, fasting fructose, triglyceride, high-density lipoprotein and cholesterol levels measured in the experimental groups’ rats. Similar to a recent report, our results revealed a significant elevation of serum triglycerides in the fructose group compared with those in the control group [8,20]. The fasting blood fructose level was also higher in fructose-fed rats, whereas the direct high-density lipoprotein level was significantly lower. Dapagliflozin administration for two weeks prevented fructose-mediated metabolic defects; the fasting fructose and triglyceride levels were significantly lower, whereas the direct high-density lipoprotein levels did not significantly differ between the fructose-dapagliflozin group and the fructose group. These results indicate that dapagliflozin suppressed fructose-induced DM.

### 2.3. Dapagliflozin Abolishes ROS Production by Inhibiting SGLT2 Expression in LECs of Rats with Fructose-Induced DM

Next, we investigated whether dapagliflozin inhibited the SGLT2 expression in LECs, thus leading to the attenuation of superoxide generation in rats with fructose-induced DM. Figure 2A shows photographs of lenses with maintained transparency; the squares under the lens are clearly visible in both the control and the fructose groups. To determine whether or not fructose-dependent ROS generation occurred in LECs of rats with fructose-induced type 2 DM, we examined the fructose effects on ROS levels. Dihydroethidium (DHE) fluorescence was used to measure superoxide levels in the LECs from rats with fructose-induced type 2 DM after eight weeks; representative images are shown in Figure 2B. DHE fluorescence was significantly increased in the LECs of fructose-fed rats compared to the control group. However, dapagliflozin reversed these effects (Figure 2B; * *p* < 0.05, *n* = 6). Real-time polymerase chain reactions (RT-PCRs) revealed significantly higher relative RNA levels of SGLT2, GLUT1, and GLUT5 in LECs or fibers of fructose-fed rats than in those of control animals; the co-administration of dapagliflozin prevented this increase (Figure 2C,D).

### 2.4. Dapagliflozin Inhibits GLUT-Induced Expression of NADPH Oxidase Subunits and RAGE in LECs from Rats with Fructose-Induced Type 2 DM

Fructose is absorbed via intestinal GLUTs, especially GLUT2 and GLUT5. To investigate whether abolishing GLUT activation in LECs decreases RAGE-induced NADPH oxidase levels in the rats with fructose-induced DM, we examined the activity and expression of GLUT1/5, RAGE, nicotinamide adenine dinucleotide phosphate reduced form oxidase 4 (NOX4) and NADPH oxidase subunits when both fructose and dapagliflozin were administered. Immunofluorescence staining demonstrated significantly higher relative expression levels of GLUT5, RAGE and NOX4 proteins in the LECs of fructose-fed rats compared to the control group; again, co-administration of dapagliflozin prevented this increase (Figure 3A,B). Similarly, immunoblotting analysis of proteins extracted from the LECs demonstrated that dapagliflozin treatment decreased the GLUT1/5 and NADPH oxidase subunit p67-phox expression in rats fed for 12 weeks with fructose (Figure 3C–E). These results indicate that dapagliflozin reduces GLUT1/5-induced RAGE expression and decreases NADPH oxidase p67-phox and NOX4 levels in the LECs of rats with fructose-induced type 2 DM.

We also extracted lenses from rats fed with 10% fructose for eight weeks with or without dapagliflozin treatment. We extracted lens tissue from rats fed with fructose for eight weeks and administered dapagliflozin ex vivo. Fluorescence imaging results indicated enhanced SGLT2, GLUT1, GLUT5 and 3-NT levels in the epithelial sections of rat lenses from the fructose-fed group. Furthermore, the administration of dapagliflozin decreased those SGLT2, 3-NT, GLUT1 and GLUT5 levels in these epithelial specimens (Figure 4A,B). Similarly, immunoblotting analysis results indicate that dapagliflozin administration decreased the RAGE, NOX2, p67, SGLT2, GLUT1 and GLUT5 protein levels in rat epithelial sections of the lens compared with the fructose group (Figure 4C,D). Based on these findings, we suggest that dapagliflozin blocked the SGLT2-induced ROS production in the LECs of type 2 DM rats.

## 3. Discussion

Epidemiological studies greatly increase the knowledge of the association between diabetes and cataract formation and defined risk factors for the development of cataracts [3]. Recent basic research emphasizes the role of ROS in disease progression. Antioxidants are beneficial for the prevention and treatment of polyol accumulation during the diabetic cataract formation in experimental studies, both in vitro and in vivo [21]. However, the exact role of ROS in the pathogenesis of diabetic cataracts is not fully understood. The studies found that increased interaction of AGEs with RAGE in the lens epithelium further increases ROS generation [22]. Activation of NADPH oxidase by AGEs mediates superoxide generation, thus leading to altered gene expression via RAGE [12]. Immunofluorescence staining demonstrates that GLUT1 and SVCT2 are highly expressed in both human and mouse LECs [17]. However, our results show that ROS levels in LEC sections were significantly higher in cataract patients independent of DM presence (Figure 1A). Interestingly, immunofluorescence staining demonstrated that the SGLT2, RAGE and GLUT1 levels in LECs were significantly elevated only in patients with DM (Figure 1). These results indicate that SGLT2 and GLUT1 may be required for RAGE-induced superoxide generation and the pathogenesis of diabetic cataract formation.

LECs also play a crucial role in the antioxidant protection of the lens. The energy required to maintain lens transparency is derived from glucose metabolism. In the lens, glucose uptake is likely either facilitated via members of the GLUT family, is sodium-dependent via members of the SGLT family, or both [23]. Lim et al. identified GLUT1 as the primary transporter that mediates glucose uptake in rat, bovine and human lenses; GLUT1 is upregulated and consequently increases glucose uptake under hyperglycemic conditions [24]. In previous studies, the presence of GLUT5 has indicated fructose transport along with glucose transport because GLUT5 has a significantly higher transport capacity for fructose than for glucose [25]. Chan et al. demonstrate that the presence of SGLT2 in the bovine ciliary body epithelium might shed light on glucose transport and the physiology of the bovine blood-aqueous barrier and glycemic control in relation to diabetic cataract formation [26]. Interestingly, our data demonstrates significantly higher relative expression levels of GLUT1 or GLUT5 proteins in the LECs of rats fed with fructose for 8 or 12 weeks, whereas the co-administration of dapagliflozin prevented these increases (Figure 2 and Figure 3).

AGEs play an essential role in degenerative lens changes; thus, approaches utilizing food with low AGE content may delay cataract formation [27]. Furthermore, the dietary-fructose-mediated generation of AGEs and activation of RAGE contribute to the metabolic syndrome [11]. DM is a global epidemic, and its prevalence is anticipated to continue to increase in correlation with the intake of fructose-sweetened beverages [28,29]. Our previous findings have extended these observations to fructose-fed rats, which show similar metabolic disturbances [16,30]. In rats, RAGE-mediated NADPH oxidase activation may participate in the regulation of fructose-induced central insulin resistance [8]. Dapagliflozin is a new therapeutic option; this SGLT2 inhibitor prevents renal glucose reabsorption, thus decreasing plasma glucose levels [31]. In the present study, dapagliflozin prevented fructose-mediated metabolic defects, and the fasting fructose and triglyceride levels were significantly lower in the fructose-dapagliflozin groups. These results indicate that dapagliflozin suppresses fructose-induced DM. Interestingly, photographs of the lenses show that the squares under the lens are clearly visible in both the normal and fructose groups, indicating that their transparency is maintained (Figure 2A). However, ROS accumulation in the LEC sections was significantly higher in fructose-fed rats than in control animals, and was reversed by dapagliflozin. Additionally, we observed that dapagliflozin might inhibit the fructose-induced superoxide generation caused by RAGE-mediated NADPH oxidase activation in LECs (Figure 4). The findings of the current study are further supported by reports that dapagliflozin use may inhibit the fructose-induced superoxide generation caused by RAGE-mediated NADPH oxidase activation, thereby preventing cataract development in patients with diabetes.

Our findings extend to GLUTs and indicate that these transporters may be required for RAGE-induced superoxide generation and the pathogenesis of diabetic cataract formation in clinical patients with DM and in a rat model of fructose-induced type 2 DM. The observed dapagliflozin effects may be mediated through the inhibition of SGLT2 and GLUT expression, thus downregulating factors such as the RAGE and NADPH oxidases, preventing ROS accumulation, and protecting the LECs (Figure 5). SGLT2 inhibitors do not lower insulin resistance or improve insulin secretion (the major pathological defects in type 2 DM) and accordingly represent a new therapeutic option [31]. The present study is the first to investigate the prophylactic benefits of SGLT2 inhibitors in diabetic cataracts. Our findings may provide further insights into the pathogenesis of diabetic cataracts; inform clinical studies investigating the association between diabetes and cataract development, and support current treatments for cataracts in patients with diabetes.

## 4. Materials and Methods

### 4.1. Reagents and Chemicals

All drugs and chemicals were obtained from Sigma-Aldrich (Sigma Chemical Co., St. Louis, MO, USA). Antibodies directed against glucose transporter 1 (GLUT1), receptor for advanced glycation end products (RAGE), Ras-related C3 botulinum toxin substrate 1 (RAC1), and sodium/glucose cotransporter 2 (SGLT2) were purchased from Abcam (Abcam, Cambridge, MA, USA; ab115730, ab3611, ab65965, and ab37296, respectively), against nicotinamide adenine dinucleotide phosphate reduced form (NADPH) oxidase 4 (NOX4) from Novus Biologicals (NB110-58849, Englewood, CO, USA), against p67-phox from Millipore (07-502, Billerica, MA, USA), and those against GLUT5 from GeneTex (GTX12098, Englewood, CO, USA).

### 4.2. Ethics Statement

The study protocols followed the Declaration of Helsinki guidelines and were independently reviewed and approved by the Institutional Review Board of the Kaohsiung Veterans General Hospital (Kaohsiung, Taiwan; IRB number: VGHKS17-CT5-10 (14 April 2017).

All data and specimens were previously collected and anonymized before analysis. Therefore, the requirement for written informed patient consent was waived by the Institutional Review Board of the Kaohsiung Veterans General Hospital. This prospective study comprised patients who underwent phacoemulsification and intraocular lens implantation between February 2016 and February 2017 at Kaohsiung Veterans General Hospital. Written informed consent was obtained from each patient and/or guardian. After receiving a full explanation of the surgical procedures and possible complications, all patients provided written informed consent. The patients were selected based upon clinically observable nuclear cataracts of grade 2 or 3 according to the Lens Opacities Classification System III [32]. Cataract surgeries were performed by Ying-Ying Chen, Tsung-Tien Wu and Ya-Hsin Kung. Exclusion criteria included cataract hardness greater than grade 3. Patients with type 1 diabetes mellitus (DM), rheumatologic disease, or any other systemic disease except type 2 DM were also excluded. The patients were classified into two groups: Patients without DM and patients with DM, but without DR. Data on DM durations and glycated hemoglobin A1c (HbA1c) levels were also evaluated. The authors appreciate the assistance from Biobank, Department of Medical Education and Research, Kaohsiung Veterans General Hospital, for processing of the clinical specimens.

### 4.3. Animals

Sixteen-week-old male WKY rats were obtained from the National Science Council Animal Facility (NSCFA) (Taipei, Taiwan) and housed in the animal facility at Kaohsiung Veterans General Hospital (VGHKS; Kaohsiung, Taiwan). NSCAF and VGHKS both have AAALACi international certification. All animals were housed in the VGHKS in an environment free of infectious organisms that are pathogenic and/or capable of interfering with the research objectives. The rats were kept in individual cages in a light- (12-h light/12-h dark cycle) and temperature-controlled (23–24 °C) room, and were given normal rat chow (Purina; St. Louis, MO, USA) and tap water ad libitum. All animal research protocols complied with the ARRIVE guidelines [33,34] and were approved by the Animal Research Committee and the Institutional Review Board at Kaohsiung Veterans General Hospital. Experiments with drug treatments were conducted in a blinded fashion. No animals were excluded from statistical analyses.

The rats were acclimated to the housing conditions for one week and then habituated to the indirect blood pressure measurement procedure for another week. After the stabilization period, the rats were randomly assigned to five groups of six rats per group with the following oral administration protocols: (1) Control group: Pure drinking water; (2) fructose group: 10% fructose in drinking water for eight weeks; (3) fructose group: 10% fructose in drinking water for 12 weeks; (4) dapagliflozin group: 10% fructose in drinking water for six weeks, followed by fructose + dapagliflozin (3 mg/kg/day) for two weeks; (5) dapagliflozin group: 10% fructose in drinking water for 10 weeks, followed by fructose + dapagliflozin for two weeks. All rats in the experimental groups developed type 2 DM with no instances of heart failure or sudden death.

### 4.4. Tissue Sample Collection

Based on the 2013 American Veterinary Medical Association (AVMA) guidelines, all rats were euthanized using 100% CO_2_. Death occurred within 2–5 min, and the lens was rapidly removed and immediately frozen on dry ice. Tissues collected from the same experimental groups were pooled and stored at −80 °C.

### 4.5. Measurement of ROS Production in Lenses from DM Patients and LECs from Type 2 DM Rats

The endogenous in vivo O_2_^¯^ levels produced in humans with DM cataracts and fructose-fed rats were determined by staining the anterior region of the lens capsule with dihydroethidium (DHE; Invitrogen, Carlsbad, CA). Lens epithelial cell (LEC)-containing slices removed from the rats were placed in OCT compound (Shandon Cryomatrix; Thermo Electron Co., Pittsburgh, PA), flash-frozen in a methylbutane-chilled bath, and then placed in liquid nitrogen. Lens capsular flaps were stained with 1 μM DHE in the dark for 20 min at 37 °C in a humidified 5% CO_2_ incubator. The samples were analyzed using fluorescence microscopy and the Zeiss LSM Image software (Carl Zeiss MicroImaging, Jena, Germany).

### 4.6. Measurement of Physiological Indices

At the final stage of the study, blood was collected from the experimental animals by cardiac puncture (the blood volume was 1–2 mL). Plasma glucose, direct high-density lipoprotein (dHDL), total cholesterol (TC), HbA1c and triglyceride (TG) levels were determined using a clinical chemistry analyzer (Ortho Clinical VITROS™ 350 System, Rochester, NY, USA). Plasma glucose and fructose levels were measured using an Ultrasensitive Rat Glucose Fructose Enzyme-Linked Immunosorbent Assay (ELISA) kit (Mercodia, Uppsala, Sweden) with a Biochrom Anthos Zenyth 200rt Microplate Reader (Cambridge, UK).

### 4.7. Blood Pressure Measurements

The systolic blood pressure of the rats was measured before the start of the fructose or dapagliflozin treatments (week 0) using a tail-cuff monitor (Noninvasive Blood Pressure System, SINGA, Taipei, Taiwan). The rats were placed in the fixer for 30 min at a constant temperature of 34 °C. During the measurement, six individual readings were obtained in rapid sequence. The highest and lowest readings were discarded, and the average of the remaining six readings was used. The systolic blood pressure of the rats was measured every day at the same time.

### 4.8. RNA Isolation and Real-Time Polymerase Chain Reactions (RT-PCR)

Total cellular RNA was isolated from the lens and fibers using an RNAqueous kit (Invitrogen). Aliquots of 1 μg RNA were reverse-transcribed using the SuperScript III First-Strand Synthesis kit (Invitrogen) according to the manufacturer’s instructions. All primers were synthesized by Operon Technologies (Alameda, CA, USA).

### 4.9. Immunoblotting Analysis

Total protein extracts were prepared by homogenizing the lenses in lysis buffer with protease and phosphatase inhibitor cocktails and then incubated for 1 h at 4 °C. The protein extracts (assessed by BCA protein assay; Pierce) were subjected to 7.5–10% SDS-Tris glycine gel electrophoresis and then transferred to a polyvinylidene difluoride membrane (GE Healthcare, Buckinghamshire, UK).

The membrane was blocked with 5% nonfat milk in Tris-buffered saline/Tween-20 buffer (10 mmol/L Tris, 150 mmol/L NaCl, and 0.1% Tween-20, pH 7.4, slightly alkali) and incubated with anti-p67-phox (07-502), anti-GLUT1 (ab115730), rat anti-GLUT5 (GTX12098) and human anti-GLUT5 (GTX83626) antibodies at 1:1000 dilutions in PBST with 5% BSA at 4 °C overnight. Peroxidase-conjugated anti-mouse or anti-rabbit secondary antibodies (1:5000) were used. The specific bands were detected with an ECL-Plus detection kit (GE Healthcare) and exposed to film. The developed films were scanned (photo scanner 4490, Epson, Long Beach, CA, USA) and analyzed with the NIH image densitometry analysis software (National Institutes of Health, Bethesda, MD, USA).

### 4.10. Immunofluorescence Stainings

Cryostat slices (10 μm) or the lens capsular flaps were incubated with anti-NOX4, anti-GLUT1, anti-GLUT5, anti-RAC1 or anti-RAGE antibodies (1:100). After being washed with phosphate-buffered saline, sections were incubated with green-fluorescent Alexa Fluor 488- or 588-conjugated donkey anti-rabbit IgG (1:200; Invitrogen) at 25 °C for 2 h. The sections were analyzed using fluorescence microscopy and Zeiss LSM Image software (Carl Zeiss MicroImaging).

### 4.11. Statistical Analysis

All statistical analyses were carried out on raw data using SPSS, version 13.0 (SPSS Inc., Chicago, IL, USA). The DM and control groups were compared using the nonparametric Mann-Whitney U test and the Chi-square test. One-way analysis of variance (ANOVA) with Scheffe’s post hoc comparison was performed to compare the group differences. Differences with *p* values < 0.05 were considered significant. Data represent the mean ± standard error of the mean (SEM) of six independent experiments, each performed in triplicate. The data and statistical analysis comply with the recommendations on experimental design and analysis in pharmacology [35].

## 5. Conclusions

In conclusion, SGLT2 inhibitors such as dapagliflozin represent a new therapeutic option for the treatment of cataracts in type 2 DM patients. The beneficial effects of dapagliflozin on the LECs may be mediated by inhibition of GLUT expression, downregulation of factors such as the RAGE and NADPH oxidases, and the prevention of ROS accumulation. These novel findings suggest that the SGLT2 inhibitor dapagliflozin may be a candidate for the pharmacological prevention of cataracts in patients with diabetes mellitus.

## Figures and Tables

**Figure 1 ijms-20-04357-f001:**
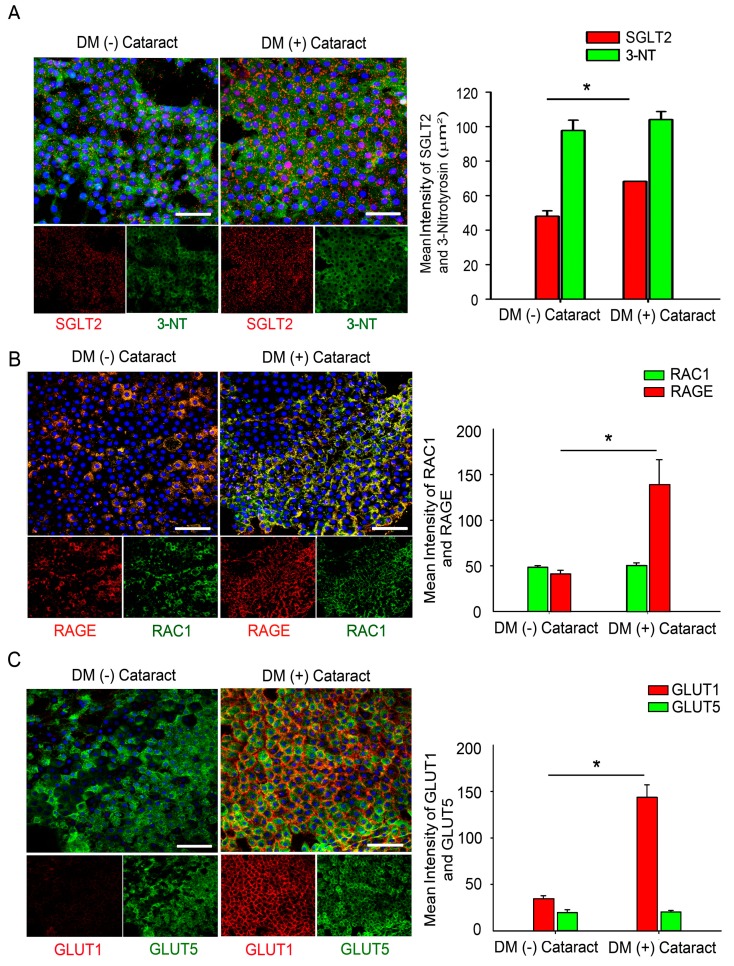
Glucose transporters (GLUTs) may act through the receptor for advanced glycation end products (RAGE) and induce Ras-related C3 botulinum toxin substrate 1 (RAC1) expression and superoxide generation in the lens tissue of cataract patients with diabetes mellitus (DM). (**A**) Representative fluorescence images of sodium/glucose cotransporter 2 (SGLT2)- and 3-NT-positive cells (green) and SGLT2-positive cells (red) in lens epithelial tissue from cataract patients with and without DM. Cell nuclei are counterstained with 4′,6-diamidino-2-phenylindole (DAPI) (blue). (**B**,**C**) Representative fluorescence images of RAGE- and GLUT1-positive cells (green) and RAC1- and GLUT5-positive cells (red) in lens epithelial tissue from cataract patients with and without DM. Cell nuclei are counterstained with DAPI (blue). Scale bar: 20 μm. Quantified values (right) are the means ± SEMs (*n* = 10 per group, separate experimental groups in each figure). * *p* < 0.05.

**Figure 2 ijms-20-04357-f002:**
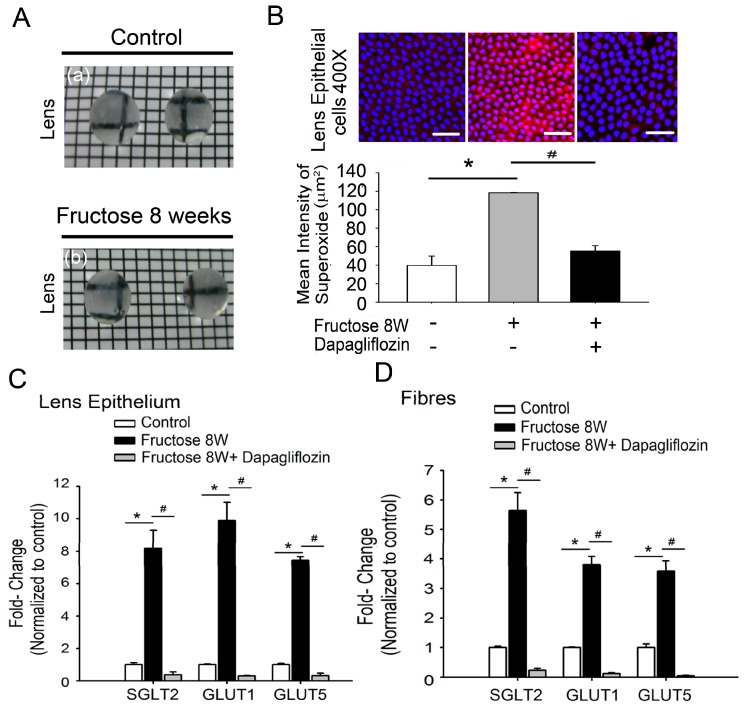
Glucose transporter (GLUT)-induced superoxide production in lenses of rats with fructose-induced type 2 diabetes mellitus (DM). (**A**) Lenses from the control (a) and fructose (b) groups. Rats exhibited fructose-induced type 2 DM after eight weeks (8 W). (**B**) Representative images of dihydroethidium-treated lens epithelial cells. Sections from rats with fructose-induced type 2 DM displayed significantly higher dihydroethidium fluorescence compared to the control group, whereas treatment with the sodium-glucose cotransporter 2 (SGLT2) inhibitor (dapagliflozin, 1.2 mg/kg/day) prevented this effect in fructose-fed rats. (**C**–**D**) Real-time polymerase chain reaction (RT-PCR) depicting SGLT2, GLUT1, and GLUT5 mRNA expression in the lens epithelium or fibers from animals with fructose-induced type 2 DM with or without dapagliflozin administration. Scale bar: 20 μm. Data are presented as the means ± SEM (*n* = 6 per group, separate experimental groups). * *p* < 0.05 vs. control; ^#^
*p* < 0.05 vs. Fructose 8 W.

**Figure 3 ijms-20-04357-f003:**
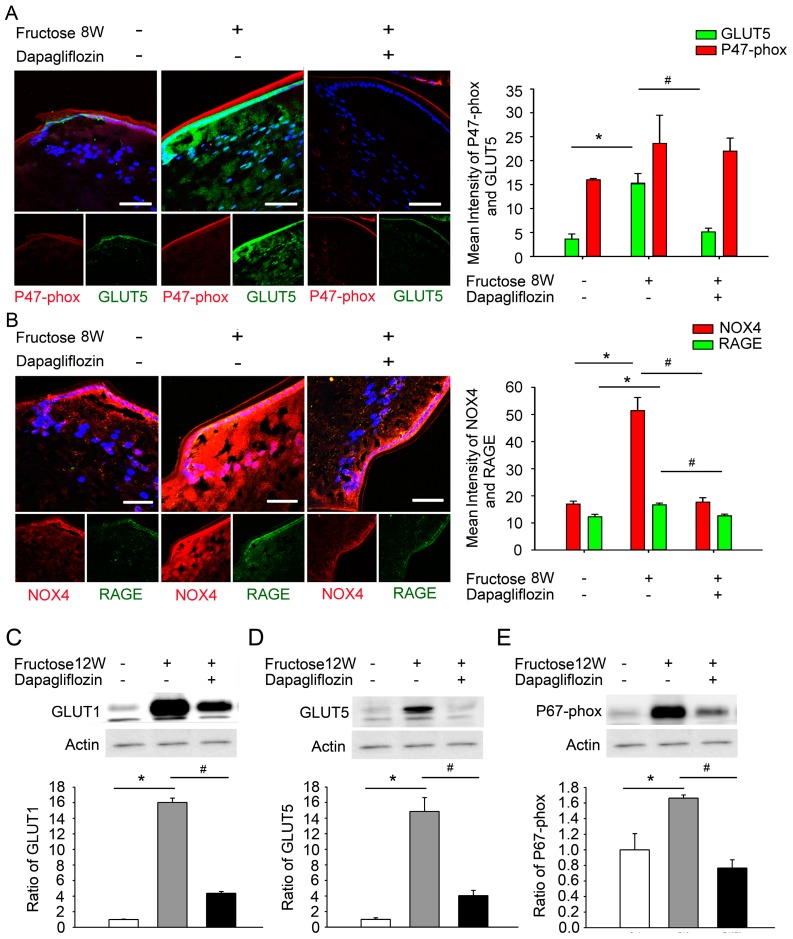
Dapagliflozin reduces glucose transporter (GLUT)-induced expression of the receptor for advanced glycation end products (RAGE), nicotinamide adenine dinucleotide phosphate reduced form oxidase 4 (NOX4) and nicotinamide adenine dinucleotide phosphate reduced form (NADPH) oxidase subunit (p67-phox) levels in the lenses of rats with fructose-induced type 2 diabetes mellitus (DM). (**A**–**B**) Representative fluorescence images of GLUT5- and RAGE-expressing cells (green) and p7-phox and NOX4-expressing cells (red) in the lens with or without a systemic administration of fructose or dapagliflozin. Cell nuclei are counterstained with DAPI (blue). (**C**–**E**) Quantitative immunoblotting analysis demonstrating that the GLUT1, GLUT5, and p67-phox levels in the lenses of rats with fructose-induced type 2 DM were significantly decreased by dapagliflozin administration. Values are presented as the means ± SEMs (*n* = 6 per group, separate experimental groups in each figure). Scale bar: 20 μm. * *p* < 0.05; ^#^
*p* < 0.05 vs. Fructose 8 W or 12 W.

**Figure 4 ijms-20-04357-f004:**
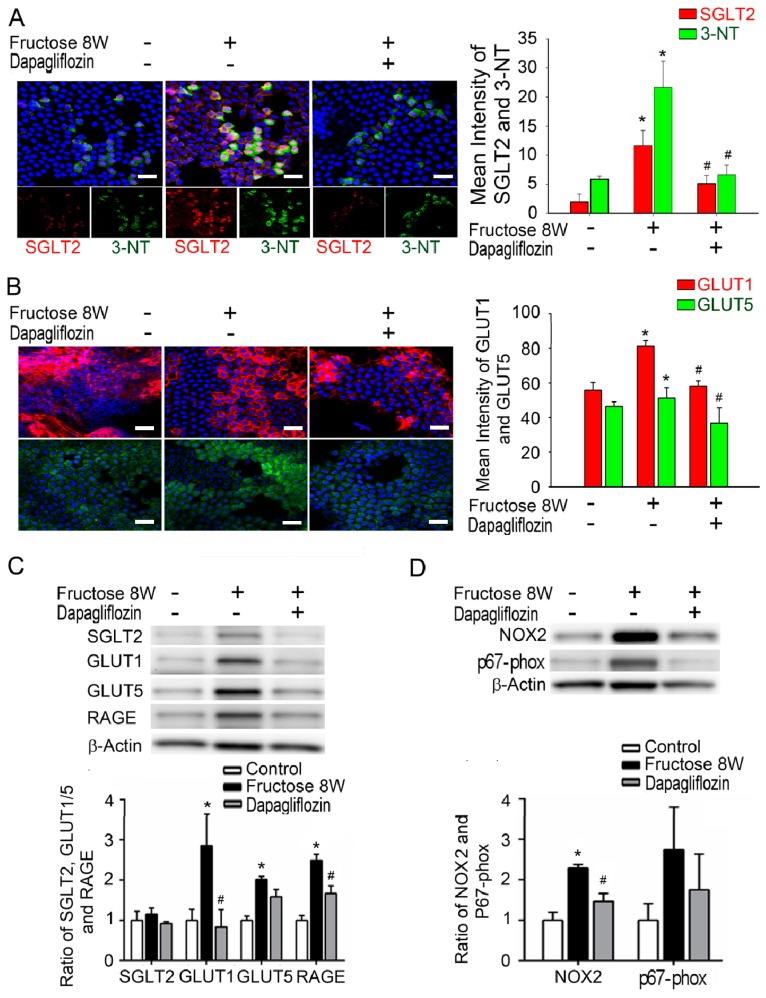
Dapagliflozin blocked the sodium-glucose cotransporter 2 (SGLT2)-induced production of the NADPH Oxidase Subunits and the receptor for advanced glycation end products (RAGE) in lens epithelial sections of type 2 diabetes mellitus (DM) rats. (**A**–**B**) Representative fluorescence images of 3-nitrotyrosine- (3-NT) and glucose transporter (GLUT)5-expressing cells (green) and SGLT2- and GLUT1-expressing cells (red) in the lens of rats with or without the systemic administration of fructose or dapagliflozin. Cell nuclei are counterstained with DAPI (blue). The presented values are the means ± SEMs (*n* = 6 per group, separate experimental groups in each figure). (**C**–**D**) Quantitative immunoblotting analysis demonstrating that the expression levels of SGLT2, GLUT1, GLUT5, RAGE, NOX2 and p67-phox in the lenses of rats with fructose-induced type 2 DM were decreased by dapagliflozin administration. Scale bar: 20 μm. The presented values are the means ± SEMs (*n* = 6 per group, separate experimental groups in each figure). * *p* < 0.05 vs. control; ^#^
*p* < 0.05 vs. Fructose 8 W.

**Figure 5 ijms-20-04357-f005:**
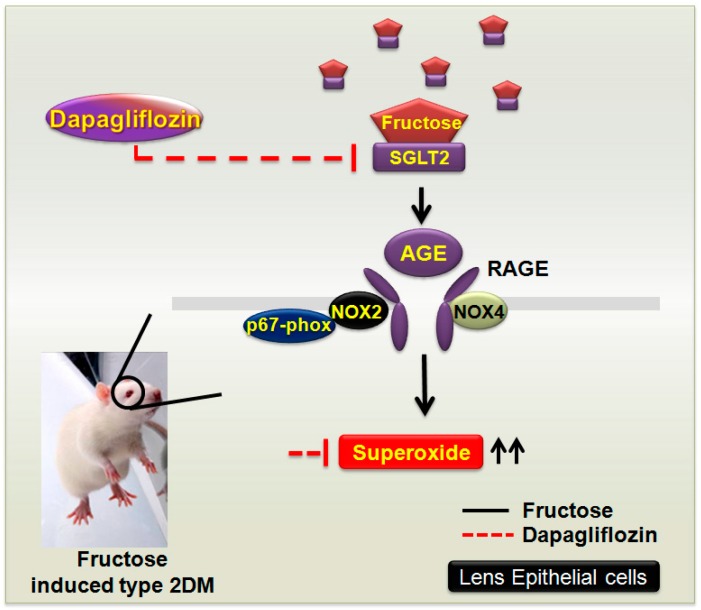
Dapagliflozin decreases the fructose-induced NOX2/4-dependent oxidative stress in the lens that is mediated by a sodium-glucose cotransporter 2 (SGLT2)-dependent mechanism in a rat model of type 2 diabetes mellitus (DM). Fructose increases the generation of reactive oxygen species by the SGLT2-induced upregulation of the expression levels for advanced glycation end products (AGE), the receptor for AGE (RAGE), and NADPH oxidase isoforms (NOX2/4) in lens epithelial cells of rats (black line). The SGLT2 inhibitor dapagliflozin acts as an important regulator of fructose by downregulating the SGLT2-induced activity of AGE-RAGE-NOX2/4 (red line).

**Table 1 ijms-20-04357-t001:** Demographics and Baseline Clinical Characteristics of the Study Participants.

	Control Group (*n* = 30)	DM Group (*n* = 30)	*p* Value
**Age (years)**	62.0 ± 9.5	62.3 ± 7.7	0.80
**Sex (male:female)**	18:12	15:15	0.34
**BMI**	19.2 ± 2.6	21.1 ± 2.9	0.90
**HbA1c**	5.50 ± 0.05	7.60 ± 1.02	0.01 *

Hemoglobin A1c (HbA1c) levels were determined in patients without diabetes mellitus (DM; control group) and in patients with DM but without diabetic retinopathy (DM group). BMI, body mass index. Values are shown as the mean ± SEM; * *p* < 0.01 vs. control group.

**Table 2 ijms-20-04357-t002:** General Characteristics of the Three Experimental Rats Groups.

Parameter	Control	Fructose 8 W	Fructose 8 W + Dapagliflozin
Systolic blood pressure (mmHg)	113.2 ± 2.5	150.1 ± 2.0 **	124.8 ± 1.0 ^††^
Fasting serum fructose (mg/dL)	2.9 ± 0.2	10.51 ± 1.0 **	4.5 ± 0.3 ^††^
dHDL (mg/dL)	50 ± 0.3	52.3 ± 0.7	50.0 ± 0.6
Triglycerides (mg/dL)	86.0 ± 0.6	256.0 ± 2.3 **	220.7 ± 6.1 ^††^
Cholesterol (mg/dL)	110.8 ± 0.7	105.3 ± 2.4	103.3 ± 0.9

All parameters were determined in rats fed with water, fructose and dapagliflozin for eight weeks (8 W) as indicated. The term dHDL refers to direct high-density lipoprotein. The values are shown as the mean ± SEM (*n* = 6 per group, separate experimental groups); ** *p* < 0.01 vs. control; ^††^
*p* < 0.01 vs. Fructose 8 W.

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
