# Peer review of "Dapagliflozin Prevents NOX- and SGLT2-Dependent Oxidative Stress in Lens Cells Exposed to Fructose-Induced Diabetes Mellitus"

_ijms, 2019, doi:10.3390/ijms20184357_

Round 1
Reviewer 1 Report
It is an interesting manuscript which is well written. Some minor issues need to be revised by the authors.
-Line 56. Reference must be added
-Line 57. Although, at the begging of the phrase the authors mention that "many studies...." at the end of the phrase (line 59) only one study is referred.
-Line 89-95. this paragraph is related to the finding of the study. Thus it should be removed and included in the conclusion part of the manuscript.
-Line 125. Please add the abbreviation for the National Science Council Animal Facility (NSCAF)
-Line 131. Although the authors are following the ARRIVE Guidelines, justification of the number of animals used in the study is missing. We are proposing either statistical analysis or use of related references.
-Line 138. In fructose group it must be clarified if 10% fructose was administered for 8 or 12 weeks.
-Line 137. Oral administration must be clarified. The administration was performed through drinking water, or with gavage.
-General comment on methodology: All animals developed DM? Any cases of failure or sudden deaths?
-We recommend authors to provide reference on the methodology used for the development of DM in rats
-Line 155. The authors provide detailed information about the blood measurements, however information about the blood collection is missing. Frequency, blood collection methodology, blood volume collected must be provided
-General comment: The human lens used were from men and women. Without underestimating the scientific importance of the study, it would be interesting to use lens from male and female rats and check if there are differences.
Author Response
-Line 56. Reference must be added
Reply:
Thank you for this comment. We have added the missing in-text reference: “Cataracts are a key factor in diabetes-associated morbidity, although the pathogenesis of diabetic cataracts is poorly understood [3]” (P2, Lines 56-58).
-Line 57. Although, at the begging of the phrase the authors mention that "many studies...." at the end of the phrase (line 59) only one study is referred.
Reply:
Thank you for your comment. We have modified the sentence from “Many studies…” to “Previous research …” (P2, Line 59).
-Line 89-95. this paragraph is related to the finding of the study. Thus it should be removed and included in the conclusion part of the manuscript.
Reply:
Thank you for your suggestion. We have moved the following sentences to the discussion section: “Our findings extend to GLUTs and indicate that these transporters may be required for RAGE-induced superoxide generation and the pathogenesis of diabetic cataract formation in clinical patients with DM and in a rat model of fructose-induced type 2 DM. The observed dapagliflozin effects may be mediated through the inhibition of SGLT2 and GLUT expression, thus downregulating factors such as the RAGE and NADPH oxidases, preventing ROS accumulation, and protecting the LECs (Fig. 5). SGLT2 inhibitors do not lower insulin resistance or improve insulin secretion (the major pathological defects in type 2 DM) and accordingly represent a new therapeutic option [1]. The present study is the first to investigate the prophylactic benefits of SGLT2 inhibitors in diabetic cataracts” (P8, Lines 262-270).
-Line 125. Please add the abbreviation for the National Science Council Animal Facility (NSCAF)
Reply:
Thank you for your comment. We have added the abbreviation for the National Science Council Animal Facility (NSCAF) to the methods section of the revised manuscript (P10, Line 309).
-Line 131. Although the authors are following the ARRIVE Guidelines, justification of the number of animals used in the study is missing. We are proposing either statistical analysis or use of related references.
Reply:
Thank you for your valuable comment. We agree with your point. In accordance with guidelines and local regulations, we intended to use the minimal number of rats to achieve an acceptable statistical power. As mentioned previously, we used 6-8 rats in each group for an experiment (repeat 3 times) based upon our lab experience, opinions from statistical experts, and published articles [2-4]. Although previously related studies supported that six animals were appropriate to achieve a meaningful statistical power, we increased the number of rats for data analysis if the condition was acceptable. In general, when a statistical result showed a trend or had marginal significance, we needed to increase the number of experimental rats to increase the statistical power. In contrast, when a result showed high significance, we considered 6-8 rats in each group for an experiment to be appropriate.
-Line 138. In fructose group it must be clarified if 10% fructose was administered for 8 or 12 weeks.
Reply:
Thank you for your comment. We have clarified our protocols thus: “After the stabilization period, the rats were randomly assigned to five groups of six rats per group with the following oral administration protocols: 1) control group: pure drinking water; 2) fructose group: 10% fructose in drinking water for 8 weeks; 3) fructose group: 10% fructose in drinking water for 12 weeks; 4) dapagliflozin group: 10% fructose in drinking water for 6 weeks, followed by fructose + dapagliflozin (3 mg/kg/day) for 2 weeks; 5) dapagliflozin group: 10% fructose in drinking water for 10 weeks, followed by fructose + dapagliflozin for 2 weeks” (P10, Lines 320-327).
-Line 137. Oral administration must be clarified. The administration was performed through drinking water, or with gavage.
Reply:
Thank you for your comment. We revised our methods section to read: “the rats were randomly assigned to five groups of six rats per group with the following oral administration protocols…” (P10, Lines 321-322).
-General comment on methodology: All animals developed DM? Any cases of failure or sudden deaths?
Reply:
Thank you for your comment. We included this wording in our methods section: “All rats in the experimental groups developed type 2 DM with no instances of heart failure or sudden death” (P10, Lines 326-327).
-We recommend authors to provide reference on the methodology used for the development of DM in rats
Reply:
Thank you for your comment. We have added the appropriate references on the methodology used for the development of DM in rats (P4, Line 128-131). For your convenience, these references are listed below:
Yeh, T. C.; Liu, C. P.; Cheng, W. H.; Chen, B. R.; Lu, P. J.; Cheng, P. W.; Ho, W. Y.; Sun, G. C.; Liou, J. C.; Tseng, C. J., Caffeine intake improves fructose-induced hypertension and insulin resistance by enhancing central insulin signaling. Hypertension 2014, 63, (3), 535-41.
Cheng, P. W.; Lin, Y. T.; Ho, W. Y.; Lu, P. J.; Chen, H. H.; Lai, C. C.; Sun, G. C.; Yeh, T. C.; Hsiao, M.; Tseng, C. J.; Liu, C. P., Fructose induced neurogenic hypertension mediated by overactivation of p38 MAPK to impair insulin signaling transduction caused central insulin resistance. Free Radic Biol Med 2017, 112, 298-307.
-Line 155. The authors provide detailed information about the blood measurements, however information about the blood collection is missing. Frequency, blood collection methodology, blood volume collected must be provided
Reply:
Thank you for your comment. We have added the pertinent information to our manuscript: “At the final stage of the study, blood was collected from the experimental animals by cardiac puncture (the blood volume was 1-2 ml)” (P10, Lines 343-344).
-General comment: The human lens used were from men and women. Without underestimating the scientific importance of the study, it would be interesting to use lens from male and female rats and check if there are differences.
Reply:
Thank you for your comment; it is a great suggestion. Once we secure funding for future studies, we will keep in mind possible gender-specific lens differences, which will certainly improve the quality of our work and help us further understand the pathogenesis and optimal treatment of DM cataracts.
References:
Chao, E. C.; Henry, R. R., SGLT2 inhibition--a novel strategy for diabetes treatment. Nat Rev Drug Discov 2010, 9, (7), 551-9. Cheng, P. W.; Lin, Y. T.; Ho, W. Y.; Lu, P. J.; Chen, H. H.; Lai, C. C.; Sun, G. C.; Yeh, T. C.; Hsiao, M.; Tseng, C. J.; Liu, C. P., Fructose induced neurogenic hypertension mediated by overactivation of p38 MAPK to impair insulin signaling transduction caused central insulin resistance. Free Radic Biol Med 2017, 112, 298-307. Yeh, T. C.; Liu, C. P.; Cheng, W. H.; Chen, B. R.; Lu, P. J.; Cheng, P. W.; Ho, W. Y.; Sun, G. C.; Liou, J. C.; Tseng, C. J., Caffeine intake improves fructose-induced hypertension and insulin resistance by enhancing central insulin signaling. Hypertension 2014, 63, (3), 535-41. Lai, C. C.; Liu, C. P.; Cheng, P. W.; Lu, P. J.; Hsiao, M.; Lu, W. H.; Sun, G. C.; Liou, J. C.; Tseng, C. J., Paricalcitol Attenuates Cardiac Fibrosis and Expression of Endothelial Cell Transition Markers in Isoproterenol-Induced Cardiomyopathic Rats. Crit Care Med 2016, 44, (9), e866-74.

Reviewer 2 Report
This paper describes the effects of dapagliflozin against fructose mediated changes in rat lens. Authors tested this in LECs from human lenses after cataract surgery and in rats. Authors performed few experiments to support this. Listed below are concerns,
What is the relevance of this study to human diabetic cataract? Present model is not an appropriate model to study diabetic cataract. Authors should use streptozotocin model in rats. How authors are sure that these samples are from patients without diabetic retinopathy? Any data to support? What is the rational to select SGLT 2 inhibitor for this study? Page 3, line 102 – check Novus Biologicals. Page 3, under 2.3 – Nothing mentioned on intake volume, how frequently they changed fructose etc. Authors are implicating this study to diabetes, fasting blood glucose should be checked in those animals. Page 4, line 169 – is it lens epithelial cell adhering to capsule or lens with epithelial cells? Page 5, 3.1 – RAGE MAY BE – sentence should be modified. Figure 1, 3 and 4 – bar graph colors are confusing – should be changed to different color for different marker. No information on - scale bar and negative control in immunofluorescences. 2A – any crosslinking in the lens? How the cells in Fig.2B was prepared, is it a transverses section or capsule adherent cells? Page 6, line 274 - no period after GLUT5. Authors missed to measure ROS in Fig.4 since the title is on ROS. Fig.4 C and D – beta actin levels are different.
Fig.5 – authors failed to measure AGE. What is the real mechanism?
Author Response
What is the relevance of this study to human diabetic cataract? Present model is not an appropriate model to study diabetic cataract. Authors should use streptozotocin model in rats. How authors are sure that these samples are from patients without diabetic retinopathy? Any data to support?
Reply:
Thank you for your comment. The first question about〝What is the relevance of this study to human diabetic cataract? Present model is not an appropriate model to study diabetic cataract. Authors should use streptozotocin model in rats.〞Investigators interested in performing small animal experiments involving diabetes mellitus (DM)-related research, including pancreatic islet transplantation studies, have a variety of methods to choose from. These methods include but are not limited to models of DM that are spontaneous or genetically-derived, chemically-induced, diet-induced, surgically-induced, and now transgenic, or knock-out derived. A single large dose of STZ is used for experiments attempting to cause severe T1DM by direct toxicity to β cells. Large doses can cause near-total destruction of β cells and little or no measurable insulin production [1]. However, at the end of 2010, the World Health Organization warned that high fructose consumption, mainly in the form of sweetened beverages, is a risk factor for several metabolic diseases [2]. Fructose-fed rats are a model of acquired systolic hypertension that displays numerous features of metabolic syndrome in humans [3]. Therefore, we used fructose-induced DM instead of ZDF rats or STZ-induced DM. The second question about〝How authors are sure that these samples are from patients without diabetic retinopathy? Any data to support?〞We highly appreciate the comment regarding diabetic retinopathy. Since all of the data and clinical specimens from Biobank were anonymized before we could access it for analysis, we did not have information regarding diabetic retinopathy status. Therefore, to avoid misrepresentation, we have deleted the sentence “any ocular comorbidity other than diabetic retinopathy (DR)” from the revised manuscript.
What is the rational to select SGLT 2 inhibitor for this study?
Reply:
Thank you for your question. We have added the rationale for SGLT 2 inhibitor use in the introduction of the revised manuscript, as seen below:
“Fructose is absorbed via glucose transporters (GLUTs) in the intestine (mainly GLUT2 and GLUT5). Recent research has focused on GLUT5 and its regulation [4]. Mantych et al. suggested that GLUT1 is the main glucose transporter and typically localized in lens epithelial cells (LECs), a part of the blood-aqueous barrier. However, GLUT5 is the main fructose transporter of the human eye [5]. Sodium-dependent glucose cotransporter 2 (SGLT2) inhibitors have recently been developed for glycemic control in patients with type 2 DM [6]. Lou et al. demonstrated that the effect of pioglitazone on insulin resistance in fructose-drinking rats correlates with AGEs/RAGE inhibition and blocking the activation of NADPH oxidase and NF kappa B [7]. According to immunofluorescence staining, GLUT1 and SGLT2 are highly expressed in both human and murine LECs [8]. LECs also play a crucial role in antioxidant protection and transport of nutrients from the aqueous humor. However, the effects of GLUTs and Rac1-dependent ROS in human LECs on the regulation of fructose-induced diabetic cataracts in vivo or in vitro are unknown. We hypothesized that the use of an SGLT inhibitor would decrease RAGE-induced NADPH oxidase expression, thereby reducing ROS generation during the formation of fructose-induced diabetic cataracts” (P2, Lines 81-94).
Page 3, line 102 – check Novus Biologicals.
Reply:
Thank you for your comment. We have added Novus Biologicals to the methods section of the revised manuscript (P9, Line 288).
Page 3, under 2.3 – Nothing mentioned on intake volume, how frequently they changed fructose etc. Authors are implicating this study to diabetes, fasting blood glucose should be checked in those animals.
Reply:
Thank you for your comment. We agree that fasting blood glucose should be checked in the experimental animals. Jeon et al. investigated that fasting blood glucose levels in 162 patients with type 2 diabetes and oral dapagliflozin decreased significantly from baseline (173.66 ± 56.62 mg / dL) to week 24 (133.09 ± 36.74 mg / dL, p < 0.05)[9]. Additionally, our previous studies provided in Table 1 (shown below) showed the fasting plasma glucose and triglyceride levels measured in rats of the experimental groups; the fasting blood glucose and triglyceride level were higher in the 4 week fructose-fed group. Dapagliflozin administration for two weeks prevented fructose-mediated metabolic defects and significantly lowered the fasting glucose and triglyceride levels. These results indicated that dapagliflozin suppressed fructose-induced DM. However, time and research fund limitations prevented us from checking fasting glucose in the current study. In the future, we will aim to check the fasting glucose in the 8 week fructose-fed group, which will improve the quality of our work.
Table 1. General Characteristics of the Three Experimental Rats Groups
|
Parameter |
Control |
Fructose 4 W |
Fructose 4 W + Dapagliflozin |
|
|
serum fasting glucose (mg/dL) |
81.2 ± 2.8 |
162.6 ± 4.2*** |
143.6 ± 5.1***†† |
|
|
serum triglyceride (mg/dL) |
82.8 ± 1.8 |
140.1 ± 13.9** |
90.1 ± 3.9 †† |
|
|
*** p<0.001 vs WKY group. |
|
|||
|
†† p<0.01 vs F4 group. |
|
|||
Page 4, line 169 – is it lens epithelial cell adhering to capsule or lens with epithelial cells?
Reply:
Thank you for your comment. The central flap of anterior lens capsular consisted of a single layer of lens epithelium with apices directed inward, and a basal laminar which forms the lens capsule isolating the lens constituents. When patients had undergone phacoemulsification, the central flap of anterior lens capsular was taken from the patients. In our studies, we take the anterior region of the lens capsule (central epithelium region shown in red below) from patients and fructose-fed rats to use for DHE and IF staining. We do not use the capsule portion with epithelial cells. We have added the description in the methods section of the revised manuscript. (P10, Lines 338)
Page 5, 3.1 – RAGE MAY BE – sentence should be modified.
Reply:
Thank you for your comment. We have revised the sentence to “GLUTs may act through RAGE and induce RAC1 expression and superoxide production in cataract lenses of DM patients” (P3 Lines 96-97).
Figure 1, 3 and 4 – bar graph colors are confusing – should be changed to different color for different marker. No information on - scale bar and negative control in immunofluorescences. 2A – any crosslinking in the lens?
Reply:
Thank you for your comments and suggestions. We have added the scale bar (20 mm) to Figures 1, 2, 3, and 4 of the revised manuscript. Besides, the antibodies selection used in immunofluorescences were following by Lim et al., studies groups publish paper in Exp Eye Res (2017) 161: 193-204. The immunofluorescences technique was performed in accordance with approved guidelines [10-12] and used the negative controls to check the specific binding in the immunofluorescences techniques.
Regarding bar graph colors, we preferred the summary results and representative fluorescence to retain the same color. For example, images of 3-nitrotyrosine (3-NT)-expressing cells (green) and SGLT2-expressing cells (red) in the lens of rats had the same green and red bar in the summary results (Figure 1A).
Besides, about 2A – any crosslinking in the lens? Yes, it has crosslinking in the lens. High ROS levels directly disturb the physiological functions of cellular macromolecules and subsequently lead to lens pacification. Currently, surgery for cataract removal and intraocular lens implants are the main treatments for diabetic cataracts. However, surgery may result in numerous severe postoperative complications, including infection, corneal edema and increased intraocular pressure, especially in elderly people and those with hyperglycemic conditions [13]. Therefore, effective therapeutic strategies must be developed for the prevention and treatment of diabetic cataracts. There is accumulating evidence that fructose promotes an oxidative imbalance by simultaneously enhancing ROS production and down-regulating key antioxidant enzymes, such as superoxide dismutase (SOD)[14, 15]. In manuscript Figure 2A shows a photograph of a lens whose transparency is maintained; the squares under the lens are clearly visible in the normal and fructose groups. To determine whether fructose-dependent ROS generation occurred in the LECs of the rats with fructose-induced type 2 DM, we examined the effects of fructose on the ROS levels. DHE fluorescence was used to estimate the superoxide levels in the LECs from rats with fructose-induced type 2 DM after 8 weeks; representative images are shown in Figure 2B. DHE fluorescence was significantly higher in the LECs taken from the fructose-fed rats than in the sections from the control groups. However, dapagliflozin reversed these effects. Therefore, the main goal of the study would be to the effect of dapagliflozin protects the lens from oxidative stress-induced damage, so the function of the lens would be an important data in the animal study to see if dapagliflozin-mediated reduction of oxidative stress can achieve such effects, which may be a potential pharmacological candidate for the prevention of cataracts in patients with diabetes.
How the cells in Fig.2B was prepared, is it a transverses section or capsule adherent cells?
Reply:
Thank you for the question. We took the lens capsule for DHE staining. We have added the description in the methods section of the revised manuscript (P10, Line 338-341).
Page 6, line 274 - no period after GLUT5. Authors missed to measure ROS in Fig.4 since the title is on ROS. Fig.4 C and D – beta actin levels are different.
Reply:
Thank you for your comments. We have revised the Figure 4 caption to: “Figure 4. Dapagliflozin blocked the sodium-glucose cotransporter 2 (SGLT2)-induced production of NADPH Oxidase Subunits and the receptor for advanced glycation end products (RAGE) in lens epithelial sections of type 2 diabetes mellitus (DM) rats” (P7, Lines 202-204).
We agree with the reviewer`s point about beta-actin levels. Therefore, we have used different methods (fluorescence and immunoblotting analysis) to estimate the SGLT2, GLUT1/5, RAGE, NOX2, and p67 levels in the rat lens epithelial sections. Immunofluorescence staining demonstrated significantly higher relative GLUT5 protein expression levels in the LECs of fructose-fed rats than in those of the control groups; co-administration of dapagliflozin prevented this increase (Figure 4A and B). The results also showed that fructose intake significantly increased SGLT2 and GLUT1/5 expression in the LECs. In contrast, dapagliflozin treatment reveres these effects (Figure 4B). Similarly, immunofluorescence staining demonstrated significantly higher relative expression levels of GLUT5, RAGE, and NOX4 proteins in LECs of fructose-fed rats in comparison to the control group; again, co-administration of dapagliflozin prevented this increase (Fig. 3A and 3B). Immunoblotting analysis of proteins extracted from LECs demonstrated that dapagliflozin treatment decreased the GLUT1/5 and NADPH oxidase subunit p67-phox expression in rats fed for 12 weeks with fructose (Fig. 3C-E). These results indicated that dapagliflozin reduces GLUT1/5-induced RAGE expression and decreases NADPH oxidase p67-phox and NOX4 levels in the LECs of rats with fructose-induced type 2 DM.
Figure 3. Dapagliflozin reduces glucose transporter (GLUT)-induced expression of the receptor for advanced glycation end products (RAGE), NOX4, and NADPH oxidase subunit (p67-phox) levels in the lenses of rats with fructose-induced type 2 diabetes mellitus (DM). (A-B) Representative fluorescence images of GLUT5- and RAGE-expressing cells (green) and p7-phox and NOX4-expressing cells (red) in the lens with or without systemic administration of fructose or dapagliflozin. Cell nuclei are counterstained with DAPI (blue). (C-E) Quantitative immunoblotting analysis demonstrating that the GLUT1, GLUT5, and p67-phox levels in the lenses of rats with fructose-induced type 2 DM were significantly decreased by dapagliflozin administration. Scale bar: 20 mm. Values are presented as the means ± SEMs (n = 6 per group, separate experimental groups in each figure). *P < 0.05; #P < 0.05 vs. Fructose 8 W or 12 W.
Figure 4. Dapagliflozin blocked the sodium-glucose cotransporter 2 (SGLT2)-induced production of NADPH Oxidase Subunits and the receptor for advanced glycation end products (RAGE) in lens epithelial sections of type 2 diabetes mellitus (DM) rats.
(A-B) Representative fluorescence images of 3-nitrotyrosine- (3-NT) and glucose transporter (GLUT)5-expressing cells (green) and SGLT2- and GLUT1-expressing cells (red) in the lens of rats with or without systemic administration of fructose or dapagliflozin. Cell nuclei are counterstained with DAPI (blue). The presented values are the means ± SEMs (n = 6 per group, separate experimental groups in each figure). (C-D) Quantitative immunoblotting analysis demonstrating that the expression levels of SGLT2, GLUT1, GLUT5, RAGE, NOX2, and p67-phox in the lenses of rats with fructose-induced type 2 DM were decreased by dapagliflozin administration. Scale bar: 20 mm. The presented values are the means ± SEMs (n = 6 per group, separate experimental groups in each figure). *P < 0.05 vs. control; #P < 0.05 vs. Fructose 8 W.
Fig.5 – authors failed to measure AGE. What is the real mechanism?
Reply:
Thank you for your comment. Hyperglycemia, a consequence of diabetes, enhances the formation of advanced glycation end products (AGEs) and senescent protein derivatives that result from the auto-oxidation of glucose and fructose [16]. In contrast, there are a number of AGE receptors, such as the advanced glycation end-product receptor (AGER) family and the scavenger receptor (SR) family, that mediate endocytosis, leading to the intracellular uptake and degradation of AGEs by their fusion with lysosomes [17]. Furthermore, AGEs peptides can be transferred to the renal system, whereas the receptors are recycled and available for further endocytosis processes [18]. The expression level and the activation of AGE receptors depend on the cell or tissue type and are regulated in response to the AGE load, other metabolic changes, and conditions such as hyperlipidemia, aging, and diabetes mellitus [19]. For example, in response to conditions with a low AGE burden, the expression of RAGE is downregulated, whereas the expression of AGER1 is upregulated. As the RAGE signaling pathway leads to the activation of transcription factors NF-κB, activator protein 1 (AP-1) and forkhead box protein O subclass (FOXO), the downregulated RAGE reduces the transcription of genes related to inflammation. The upregulation of AGER1 also inhibits these transcription factors through the sirtuin-1 (SIRT1) pathway. In addition, the increased expression of AGER1 may accelerate the intracellular degradation of AGEs, which results in an overall lower degree of inflammation caused by AGEs. However, when there is an AGE burden, RAGE is upregulated, leading to increased inflammation. The prolonged high AGE burden leads to the downregulation of AGER1, which then cannot exert strong inhibitory effects on RAGE signaling or reduce the levels of AGEs by their degradation [20]. Specifically, the AGE–RAGE interaction stimulates Janus kinase/signal transducers and activators of transcription (JAK/STAT), p38 mitogen-activated protein kinase (p38 MAPK), extracellular signal-regulated protein kinases 1 and 2 (ERK 1/2), and c-Jun N-terminal kinase (JNK), which leads to the activation of transcription factors NF-κB and interferon-stimulated response elements (ISRE). This causes increased expression of cytokines, growth factors, and adhesion molecules. However, the AGE–RAGE interaction directly induces the generation of ROS via NADPH oxidases and/or other previously characterized mechanisms [21]. Furthermore, the dietary-fructose-mediated generation of AGEs and the activation of RAGE contributes to metabolic syndrome [22]. In particular, most studies have found that the increased generation of RAGE in the brains of fructose-fed rats may contribute to the impairment of insulin resistance in the brain [7, 23]. Conversely, following caffeine (caffeine’s antioxidant ability protects membranes against oxidative damage) treatment, caffeine displayed a significant decrease in RAGE expression compared to the fructose rats [15]. Furthermore, Lou et al. demonstrated that the effect of pioglitazone on insulin resistance in fructose-drinking rats correlates with AGE/RAGE inhibition and blocking of NADPH oxidase and NF kappa B activation [7]. Therefore, our studies of LECs also demonstrated that dapagliflozin downregulates RAGE-induced NADPH oxidase expression via inactivation of glucose transporters and reduction in ROS generation. These novel findings suggest that the SGLT2 inhibitor dapagliflozin may be a candidate for the pharmacological prevention of cataracts in patients with DM.
References
Hayashi, K.; Kojima, R.; Ito, M., Strain differences in the diabetogenic activity of streptozotocin in mice. Biol Pharm Bull 2006, 29, (6), 1110-9. Aller, E. E.; Abete, I.; Astrup, A.; Martinez, J. A.; van Baak, M. A., Starches, sugars and obesity. Nutrients 2011, 3, (3), 341-69. Tran, L. T.; Yuen, V. G.; McNeill, J. H., The fructose-fed rat: a review on the mechanisms of fructose-induced insulin resistance and hypertension. Mol Cell Biochem 2009, 332, (1-2), 145-59. Wilder-Smith, C. H.; Li, X.; Ho, S. S.; Leong, S. M.; Wong, R. K.; Koay, E. S.; Ferraris, R. P., Fructose transporters GLUT5 and GLUT2 expression in adult patients with fructose intolerance. United European Gastroenterol J 2014, 2, (1), 14-21. Mantych, G. J.; Hageman, G. S.; Devaskar, S. U., Characterization of glucose transporter isoforms in the adult and developing human eye. Endocrinology 1993, 133, (2), 600-7. Nauck, M. A., Update on developments with SGLT2 inhibitors in the management of type 2 diabetes. Drug Des Devel Ther 2014, 8, 1335-80. Liu, X.; Luo, D.; Zheng, M.; Hao, Y.; Hou, L.; Zhang, S., Effect of pioglitazone on insulin resistance in fructose-drinking rats correlates with AGEs/RAGE inhibition and block of NADPH oxidase and NF kappa B activation. Eur J Pharmacol 2010, 629, (1-3), 153-8. Ma, N.; Siegfried, C.; Kubota, M.; Huang, J.; Liu, Y.; Liu, M.; Dana, B.; Huang, A.; Beebe, D.; Yan, H.; Shui, Y. B., Expression Profiling of Ascorbic Acid-Related Transporters in Human and Mouse Eyes. Invest Ophthalmol Vis Sci 2016, 57, (7), 3440-50. Jeon, H. J.; Ku, E. J.; Oh, T. K., Dapagliflozin improves blood glucose in diabetes on triple oral hypoglycemic agents having inadequate glucose control. Diabetes Res Clin Pract 2018, 142, 188-194. Lim, J. C.; Perwick, R. D.; Li, B.; Donaldson, P. J., Comparison of the expression and spatial localization of glucose transporters in the rat, bovine and human lens. Exp Eye Res 2017, 161, 193-204. Hewitt, S. M.; Baskin, D. G.; Frevert, C. W.; Stahl, W. L.; Rosa-Molinar, E., Controls for immunohistochemistry: the Histochemical Society's standards of practice for validation of immunohistochemical assays. J Histochem Cytochem 2014, 62, (10), 693-7. Ghauharali, R. I.; Brakenhoff, G. J., Fluorescence photobleaching-based image standardization for fluorescence microscopy. J Microsc 2000, 198 (Pt 2), 88-100. Kumar, P. A.; Reddy, P. Y.; Srinivas, P. N.; Reddy, G. B., Delay of diabetic cataract in rats by the antiglycating potential of cumin through modulation of alpha-crystallin chaperone activity. J Nutr Biochem 2009, 20, (7), 553-62. Francini, F.; Castro, M. C.; Schinella, G.; Garcia, M. E.; Maiztegui, B.; Raschia, M. A.; Gagliardino, J. J.; Massa, M. L., Changes induced by a fructose-rich diet on hepatic metabolism and the antioxidant system. Life Sci 2010, 86, (25-26), 965-71. Yeh, T. C.; Liu, C. P.; Cheng, W. H.; Chen, B. R.; Lu, P. J.; Cheng, P. W.; Ho, W. Y.; Sun, G. C.; Liou, J. C.; Tseng, C. J., Caffeine intake improves fructose-induced hypertension and insulin resistance by enhancing central insulin signaling. Hypertension 2014, 63, (3), 535-41. Guglielmotto, M.; Aragno, M.; Tamagno, E.; Vercellinatto, I.; Visentin, S.; Medana, C.; Catalano, M. G.; Smith, M. A.; Perry, G.; Danni, O.; Boccuzzi, G.; Tabaton, M., AGEs/RAGE complex upregulates BACE1 via NF-kappaB pathway activation. Neurobiol Aging 2012, 33, (1), 196 e13-27. Ohgami, N.; Nagai, R.; Miyazaki, A.; Ikemoto, M.; Arai, H.; Horiuchi, S.; Nakayama, H., Scavenger receptor class B type I-mediated reverse cholesterol transport is inhibited by advanced glycation end products. J Biol Chem 2001, 276, (16), 13348-55. Ott, C.; Jacobs, K.; Haucke, E.; Navarrete Santos, A.; Grune, T.; Simm, A., Role of advanced glycation end products in cellular signaling. Redox Biol 2014, 2, 411-29. Vlassara, H., The AGE-receptor in the pathogenesis of diabetic complications. Diabetes Metab Res Rev 2001, 17, (6), 436-43. Poulsen, M. W.; Hedegaard, R. V.; Andersen, J. M.; de Courten, B.; Bugel, S.; Nielsen, J.; Skibsted, L. H.; Dragsted, L. O., Advanced glycation endproducts in food and their effects on health. Food Chem Toxicol 2013, 60, 10-37. Wautier, M. P.; Chappey, O.; Corda, S.; Stern, D. M.; Schmidt, A. M.; Wautier, J. L., Activation of NADPH oxidase by AGE links oxidant stress to altered gene expression via RAGE. Am J Physiol Endocrinol Metab 2001, 280, (5), E685-94. Miller, A.; Adeli, K., Dietary fructose and the metabolic syndrome. Curr Opin Gastroenterol 2008, 24, (2), 204-9. Coughlan, M. T.; Thorburn, D. R.; Penfold, S. A.; Laskowski, A.; Harcourt, B. E.; Sourris, K. C.; Tan, A. L.; Fukami, K.; Thallas-Bonke, V.; Nawroth, P. P.; Brownlee, M.; Bierhaus, A.; Cooper, M. E.; Forbes, J. M., RAGE-induced cytosolic ROS promote mitochondrial superoxide generation in diabetes. J Am Soc Nephrol 2009, 20, (4), 742-52.

Round 2
Reviewer 2 Report
NA